# Coexistence of resistance oscillations and the anomalous metal phase in a lithium intercalated TiSe₂ superconductor

Menghan Liao[1], Heng Wang[1], Yuying Zhu[1,2], Runan Shang[2], Mohsin Rafique [1], Lexian Yang[1,3], Hao Zhang [1,2,3], Ding Zhang [1,2,3,4 ✉] & Qi-Kun Xue[1,2,3,5 ✉]

Superconductivity and charge density wave (CDW) appear in the phase diagram of a variety of materials including the high-$T_c$ cuprate family and many transition metal dichalcogenides (TMDs). Their interplay may give rise to exotic quantum phenomena. Here, we show that superconducting arrays can spontaneously form in TiSe₂–a TMD with coexisting superconductivity and CDW—after lithium ion intercalation. We induce a superconducting dome in the phase diagram of Li$_x$TiSe₂ by using the ionic solid-state gating technique. Around optimal doping, we observe magnetoresistance oscillations, indicating the emergence of periodically arranged domains. In the same temperature, magnetic field and carrier density regime where the resistance oscillations occur, we observe signatures for the anomalous metal—a state with a resistance plateau across a wide temperature range below the superconducting transition. Our study not only sheds further insight into the mechanism for the periodic electronic structure, but also reveals the interplay between the anomalous metal and superconducting fluctuations.

[1] State Key Laboratory of Low Dimensional Quantum Physics and Department of Physics, Tsinghua University, Beijing, China. [2] Beijing Academy of Quantum Information Sciences, Beijing, China. [3] Frontier Science Center for Quantum Information, Beijing, China. [4] RIKEN Center for Emergent Matter Science (CEMS), Wako, Saitama, Japan. [5] Southern University of Science and Technology, Shenzhen, China. ✉email: dingzhang@mail.tsinghua.edu.cn; qkxue@mail.tsinghua.edu.cn

Titanium diselenide is an archetypical material system in which strong electron correlations help engender a variety of intriguing quantum phases[1–9]. At room temperature, TiSe$_2$ is a semi-metal with hole bands at the Γ point and electron pockets at the $L$ point[10,11]. By lowering $T$ to around 200 K, the CDW order—with a signature of exciton condensation[7]—emerges in bulk TiSe$_2$[1,3,8]. At $T = 50$ K, a gyrotropic electronic order has been recently realized by cooling TiSe$_2$ down from 250 K, while shining circularly polarized light[9]. At a still lower temperature scale, a superconducting dome—with a maximal transition temperature $T_{c0}$ ranging from 1.8 to 4.2 K—can be realized via chemical intercalation or applying pressure to the bulk material of TiSe$_2$[2,4,12,13]. Lately, such a superconducting dome has also been observed in TiSe$_2$ in the two-dimensional (2D) limit by using the ionic liquid gating technique[6]. Of particular interest, are the peculiar magnetoresistance oscillations in such an ultrathin superconductor. It indicates that a spatially periodic superconducting structure can form spontaneously. Although a spatially modulated order has been seen in the superconducting state of high-$T_c$ cuprates[14], its period falls on the nanometer scale. In contrast, the domain size in liquid-gated TiSe$_2$ is as large as a few hundred nanometers. It clearly demands a distinct type of competing interactions. However, further progress in understanding the origin of the large periodic structure remains slow. One of the limiting factors is that ultrathin TiSe$_2$ under liquid gating has remained as the only platform showing such an exotic behavior. Here we report that TiSe$_2$ with a thickness of 50 nm, upon lithium intercalation, becomes a superconductor that shows magnetoresistance oscillations below $T_{c0}$. More importantly, we observe distinctly different doping and temperature dependences than those in the ultrathin case, indicating a broader scope for these spontaneously ordered electronic structures. Interestingly, in the exact parameter space of carrier density, magnetic field and temperature ($n$-$B$-$T$) for the resistance oscillations, we also observe the anomalous metal (AM) states. Our study suggests that periodically arranged domains may contribute to the emergence of the AM.

## Results

**Electric field-controlled lithium intercalation.** Our samples are thick TiSe$_2$ flakes (50 nm) and we intercalate lithium ions via the substrate—a solid ion conductor[15–17]. Figure 1a illustrates the sample and the measurement scheme. At room temperature, positive back-gating above a threshold voltage induces lithium ion intercalation. The presence of lithium ions in TiSe$_2$ can be demonstrated by element-sensitive mass spectroscopy (Supplementary Note 1). Lithium ions preferably go into the inter-lamellar gaps of the transition metal dichalcogenide material[13,15,16,18,19], as shown in Fig. 1a. This intercalation expands the lattice in the c-axis, which is captured by our in situ atomic force microscope (AFM) study shown in Fig. 1b. The height from the top of TiSe$_2$ to the substrate surface increases from about 80 to about 100 nm after an extended period of gating (we estimate a c-axis expansion of about 12% for samples S1 and S2 studied by low-temperature transport, see Supplementary Fig. 2). We discuss the dynamic process of the intercalation in Supplementary Note 1.

The amount of injected lithium ions can be controlled by the gating period of time (inset in Fig. 1c), as the intercalation can be terminated by cooling down to 250 K. Transport properties at low temperatures can be subsequently studied. Figure 1c collects the temperature-dependent resistance curves, showing a dramatic decrease of resistance after consecutive gating. The first few curves all exhibit a prominent resistance peak at around 160 K. This resistance anomaly has been widely attributed to the partially gapped Fermi surface and the up-shifting of chemical

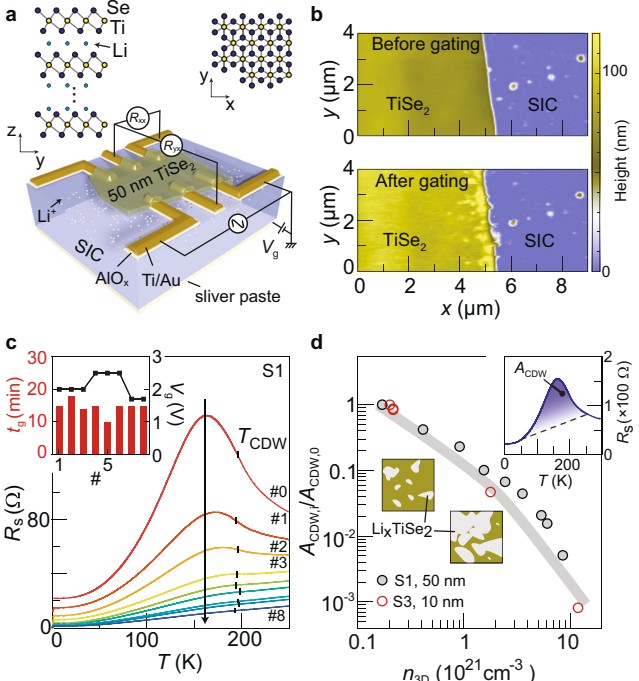

**Fig. 1 Electric field-controlled lithium intercalation and transport characterization. a** Schematic drawing of a thick TiSe$_2$ flake on the solid ionic conductor (SIC) with the pre-patterned electrodes. The electrodes consist of Ti/Au (5 nm/30 nm) on top of a thin layer of AlO$_x$ (10 nm). This oxide layer prevents lithium intercalation into the contacts. The upper panels show the stick-ball model of the TiSe$_2$ structure. **b** AFM images taken before (top) and after (bottom) gating. The gating was performed in situ on the AFM platform (~3 V, 1 h). Details are given in the Supplementary Information. **c** Sheet resistance of sample S1 (50 nm thick) as a function of temperature at consecutive gating stages. The vertical bars at about 200 K mark the local minima/kink in $dR_s/dT$ (see Supplementary Fig. 12), which we use to define $T_{CDW}$[1]. The arrow indicates the gating sequence as explained by the inset. **d** Comparison between sample S1 (50 nm) and sample S3 (10 nm) on the suppression of resistance peak related to CDW. To quantify the peak magnitude, we use the area between each curve and a dashed line connecting two points at 50 and 250 K, indicated as $A_{CDW}$ in the inset. The main panel shows the normalized area (divided by the area for the ungated stage) as a function of three-dimensional (3D) carrier density.

potential caused by the CDW[2,6,12,20,21], such that the position where $dR_s/dT$ has a local minimum correspond to the onset temperature of CDW—$T_{CDW}$[1] (Supplementary Fig. 12). It is worth noting that our Li$_x$TiSe$_2$ behaves differently from Ti$_{1+x}$Se$_2$[1], Cu$_x$TiSe$_2$[2,3], or ultrathin TiSe$_2$ under ionic liquid gating[6], because here $T_{CDW}$ stays unchanged while the peak magnitude becomes smaller upon doping. A similar behavior was observed previously in lithium-intercalated TaS$_2$[22]. It could be an indication that Li$^+$ ions fail to penetrate the complete sample. For example, the top few layers may not be reached by Li$^+$ ions and they show the CDW transition at the original value. However, if Li$^+$ ions only penetrate vertically to the few layers on the bottom, it would not be able to generate the large thickness increase observed experimentally (Fig. 1b). Moreover, the magnitude of the resistance peak decreases with the carrier density per unit volume (Fig. 1d), irrespective of thickness variations (here, the carrier densities are determined from Hall measurements at 1.6 K, Supplementary Note 3). It again confirms that lithium intercalation can affect the complete sample in the vertical direction. We speculate that the doped regions with suppressed CDW percolate

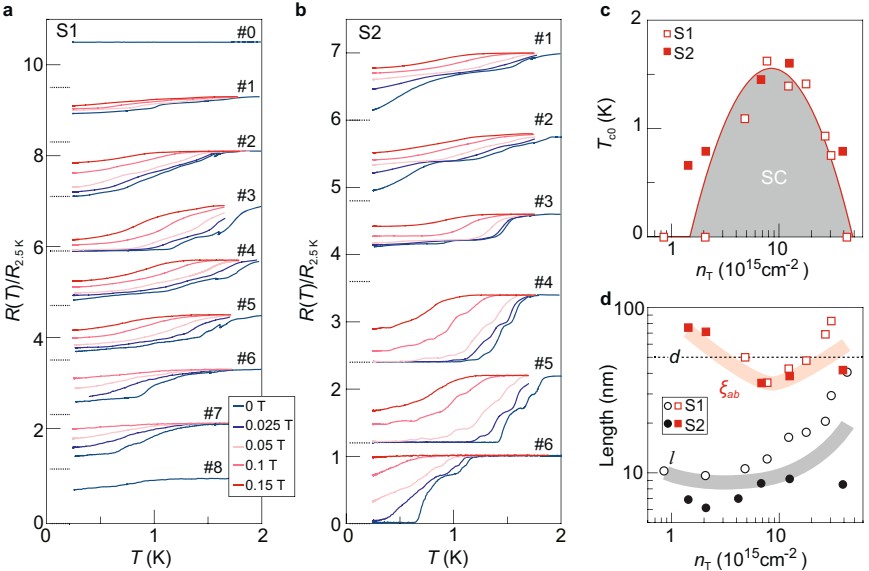

**Fig. 2 Superconductivity in lithium-intercalated TiSe₂. a, b** Normalized resistances at low temperatures for samples S1 and S2 at different gating stages and at a set of perpendicular magnetic fields. Data are vertically offset for clarity. The dotted lines mark the positions of zero resistances for the curves at each gated state. **c** Phase diagram of superconductivity in Li$_x$TiSe₂. $T_{c0}$ is defined as the temperature where the resistance drops to 50% $R_n$ at zero magnetic field. **d** Comparison between the mean free path $l$ (circles) and superconducting coherence length $\xi$ (squares) as a function of carrier density of the two samples (S1 and S2). Here, $\xi$ values are obtained by using the 50% $R_n$ criterion to define $T_c$. The dotted line marks the sample thickness for both S1 and S2.

in the plane, as sketched in the inset of Fig. 1d. Concomitantly, the regions not intercalated by lithium ions shrink with further doping but still possess the original $T_{CDW}$.

**Superconducting properties of Li$_x$TiSe₂.** Intercalating Li⁺ ions into TiSe₂ gives rise to superconductivity. Figure 2a, b show the resistance curves at low temperatures for two samples as a function of gating. A superconducting transition emerges with increasing doping and zero-resistance state develops at low temperatures at the optimal doping. The superconductivity then becomes weaker at higher doping. At each gated state, we further measure the temperature-dependent resistance at a set of magnetic fields that are applied perpendicular to the sample plane. With increasing magnetic field, the superconductor transitions back to a normal conductor. Figure 2c summarizes the dome-like behavior of the superconducting transition temperature as a function of the total carrier density $T_{c,0}(n_T)$. Here, $n_T$ is determined from the Hall measurement at 1.6 K (Supplementary Fig. 4). The total injected charge carriers amount to $4.2 \times 10^{16}$ cm$^{-2}$, far exceeding the upper limit of electrostatic gating by using ionic liquid[6]. From the magnetic field response, we further estimate the in-plane coherence length (Fig. 2d and Supplementary Fig. 7). $\xi_{ab}$ becomes smaller than the sample thickness $d$ around the optimal doping. If taking into account the anisotropy effect of this layered superconductor, the out-of-plane coherence length always satisfies: $\xi_c < d$, different from that of ultrathin superconductors where $d < \xi_c$[23–28]. The angular dependence of $B_{c2}$ (shown in Supplementary Fig. 3 for a sample in the underdoped regime) follows closely the 2D Tinkham formula rather than an anisotropic Ginzburg–Landau model. The corresponding superconducting thickness $d_{SC}$ is determined to be about 20 nm, smaller than $d$ but much thicker than a single atomic layer of TiSe₂. Therefore, our sample may consist of a stack of 2D superconductors that are weakly coupled in the vertical direction. By taking into account this effect, we estimate the mean free path $l$ of each layer based on the Hall density and the sheet resistance at 1.6 K (Supplementary Note 3). Figure 2d shows that $\xi_{ab} > l$ at

different gating stages, indicating that Li$_x$TiSe₂ superconductor is in the dirty limit.

**Magnetoresistance oscillations.** We further investigate the superconductor by sweeping the perpendicular magnetic field at fixed temperatures (Fig. 3a–c). Notably, small resistance spikes occur (marked by arrows) at certain doping levels in Fig. 3a, b. They occur in the transition regime from the superconducting state to the normal state and locate symmetrically at positive and negative magnetic fields. These peaks are independent of the sweeping rates (Supplementary Fig. 13). Angular-dependent magnetic field study further reveals that the oscillations only depend on the perpendicular component of the magnetic field (Supplementary Fig. 14). The periodicity $\Delta B$ of the oscillations can be identified from the Fourier transform (Fig. 3d, e and Supplementary Fig. 15).

Similar oscillations were observed before in ultrathin TiSe₂ under ionic liquid gating[6]. They were attributed to the Little–Parks effect caused by the spontaneous formation of periodic superconducting loops. The periodic structure consisted of domains of commensurate CDW (CCDW) with incommensurate CDW (ICDW) in the domain boundaries as a result of the competition between the ICDW and CCDW. At low temperatures, the ICDW regions became superconducting, while the CCDW regions remained non-superconducting. In fact, correlation between the CDW and the resistance oscillations can be seen in our doping-dependent study. Figure 3c shows that the resistance spikes disappear in the over-doped regime and the Fourier transform shows no clear features (the remnant multi-step behavior in Fig. 3c reflects the existence of multiple superconducting transitions.). The over-doped regime is the one where the peak in the temperature-dependent resistance becomes hardly visible (Fig. 1c). Hall measurements further reveal that the electronic condition favored by CDW is no longer satisfied (Supplementary Fig. 19). In general, CDW gets suppressed in the over-doped regime and this behavior is correlated with the disappearance of resistance oscillations. We therefore conclude

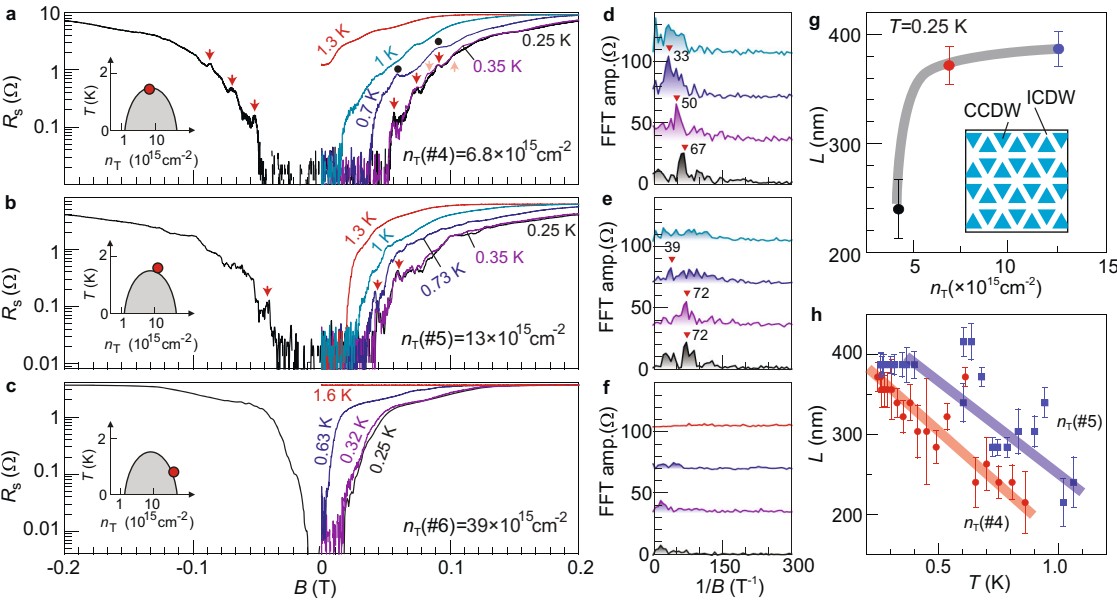

**Fig. 3 Temperature and doping dependences of the magnetoresistance oscillations in lithiated TiSe₂. a–c** Magnetoresistances of sample S2 at selected temperatures. The red and pink arrows mark the peak positions of quantum oscillations. Left insets indicate the doping level in the schematic phase diagram. **d–f** Fast-Fourier Transforms (FFT) of magnetoresistance curves at 0.25, 0.35, and 0.7 K (**a**)/0.73 K (**b**), and 1 K (**a**, **b**)/1.6 K (**c**), respectively. The red triangles with numbers mark the peak positions of the FFT data. **g**, **h** Doping and temperature dependences of the domain size $L$. Error bars are calculated from the full-width at half maximum (FWHM) of the FFT peaks. Inset to **g** schematically illustrates the periodic domain structure.

that a similar CDW domain structure as that reported in the ultrathin case emerges in our system—at around optimal doping, which gives rise to the resistance oscillations.

Our work demonstrates a spontaneously emerged periodic domain structure even in a thick sample of TiSe₂. By using $\Delta B$ evaluated from FFT, we extract the domain size $L$ at different doping levels and at different temperatures, as it follows the equation: $\Delta B \cdot L^2 = h/2e$[6]. Interestingly, both the doping and the temperature dependences of $L$ are distinctly different from those seen in ultrathin TiSe₂, shedding further insight into this exotic phenomenon. In ref. [6], $L$ quickly decreased when the carrier density increased. In comparison, Fig. 3g shows that $L$ increases with doping. This increasing trend of $L$ with doping is further supported by data in Fig. 3h, where $L$ at higher doping is systematically larger than that at lower doping across a wide temperature window. The decreasing $L$ at higher temperature is also different from the temperature-independent behavior reported before. We speculate that ICDW in Li$_x$TiSe₂ in the optimal to over-doped regime may possess an onset temperature close to $T_{c0}$, such that the periodic structure becomes sensitive to small temperature variations.

**Anomalous metal phase.** Another interesting feature in our transport data is the resistance plateau at low temperature once a small magnetic field is applied, as can be seen in the traces of Fig. 2 already. In Fig. 4a, we plot the resistance in the logarithmic scale as a function of the inverse temperature to better highlight this phenomenon. Taking the trace at 0.075 T in the left panel of Fig. 4a, it exhibits the expected thermally activated behavior from around $1/T = 0.72\,K^{-1}$ (corresponding to 1.39 K) but then shows deviation at around $1/T = 1.24\,K^{-1}$ (0.81 K, empty circle). A resistance plateau with a value about 0.37 Ω, i.e., 4% of the normal state resistance $R_n$, develops from $1/T = 2.5-4\,K^{-1}$ (0.4–0.25 K). Insufficient cooling can be immediately ruled out, because otherwise the same technical issue should be present at different doping levels. In the over-doped regime (right panel of Fig. 4a), however, the resistance plateau is absent. Instead, the resistance

shows a continuously decreasing trend down to the base temperature. The data sets here are all obtained after the installation of radio frequency filters (Supplementary Fig. 8), ruling out the extrinsic perturbation effect at radiofrequencies[29]. We further note that the resistance plateau cannot be explained by a classical percolation model, in which the superconducting regions are separated by the normal state regions. To account for the remnant resistance of 4% $R_n$ for the trace at 0.075 T in the left panel of Fig. 4a, the fraction of normal state regions has a maximal length of 200 nm, because the distance between the two voltage probes for measuring the resistance is 5 μm. On the other hand, the normal state coherence length $\xi_n = \sqrt{\hbar D/k_B T}$ exceeds 200 nm at $T < 0.3$ K (here, $D = v_F l/2$ is the diffusion constant and $v_F$ is the Fermi velocity. We estimate $v_F$ to be $3 \times 10^5\,m\,s^{-1}$ based on the band mass reported in ref. [8]). In other words, the normal state region can be fully proximitized by the superconducting region such that the resistance should drop down to zero at $T < 0.3$ K.

In fact, a resistance plateau across a wide temperature range below the superconducting transition constitutes the key characteristic of the so-called AM[23–32]. The origin of the AM has been under intensive debate[29,33]. The resistance plateau indicates the persistence of some dissipation. One of the promising theoretical models of AM attributes such dissipation to quantum creeping of vortices. In this model, the magnetoresistance of the AM is predicted to host an exponential increase. Supplementary Fig. 18 shows that the magnetoresistance in the AM phase of our Li$_x$TiSe₂ indeed follows this behavior. However, the AM is widely believed to be related to strong superconducting fluctuations in 2D, which separate the system into superconducting puddles and normal regions. As shown in Fig. 4b, the AM indeed occurs in a lot of ultrathin 2D superconductors. By contrast, our sample is much thicker ($d = 50$ nm) and consists of a stack of superconducting layers. The extracted superconducting thickness of $d_{SC} = 20$ nm is also one order of magnitude larger than the typical value in those ultrathin systems. For instance, $d_{SC} = 1.8$ nm for ionic liquid-gated

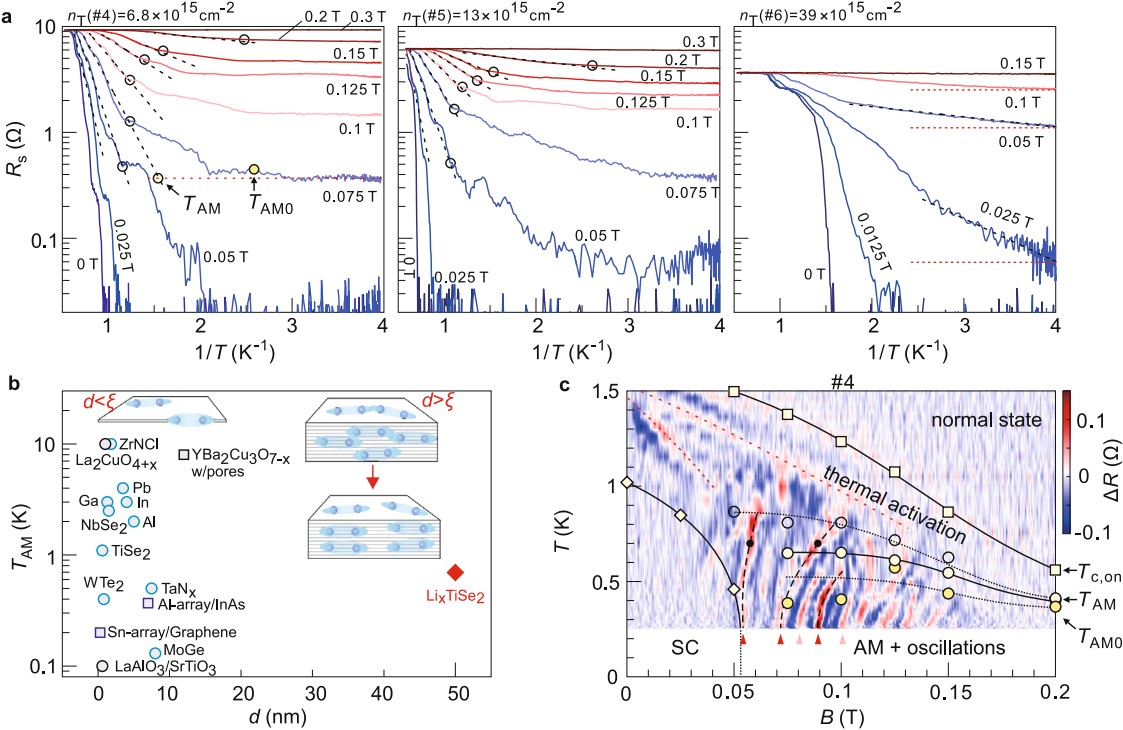

**Fig. 4 Anomalous metal phase and its coexistence with the resistance oscillations in Li$_x$TiSe$_2$. a** Arrhenius plots of sheet resistances of sample S2 at selected magnetic fields at three doping levels. Dashed lines are linear fits. Horizontal lines in the right panel are a guide to the eye. Empty circles mark the temperature points at which $R_s$ deviates from the dashed lines by 5% in decreasing $T$ (increasing $1/T$). The trace at 0.075 T in the left panel helps illustrate the method for further defining the anomalous metal. The dotted horizontal line represents the resistance plateau, which is obtained by averaging $R_s$ from 0.25 to 0.35 K. The SD of $R_s$ in this temperature window is $\delta R_s$. $T_{AM}$ marks the crossing point between the dashed and horizontal lines. $T_{AM0}$ marks the point at which $R_s$ first exceeds the horizontal line by $10\delta R_s$ when $T$ increases from 0.25 K. **b** Overview of the onset temperature of AM as a function of the thickness of superconducting systems. The previously reported systems include the following: elemental metals[23] and amorphous films[24,25], transition metal dichalcogenides[27,28,34], oxides[31,36,37], superconducting arrays[30–32], and ZrNCl[26]. The red diamond indicates Li$_x$TiSe$_2$ of this work. Left (right) inset illustrates the situation where the film thickness $d$ is smaller (larger) than the superconducting coherence length $\xi$. The inset on the right further illustrates the possible alignment of the superconducting puddles. **c** Temperature and magnetic field dependence of the quantum oscillations at $n_T(\#4) = 6.8 \times 10^{15}$ cm$^{-2}$ (sample S2). A smoothed background, which is obtained by running a moving average over data points within 9.2 mT, has been subtracted from each of the original magnetoresistance trace. The circles are corresponding temperature points extracted from **a**. The squares/diamonds mark the temperature points where the resistance drops to 90% or 0.4% of the normal state resistance, respectively. The dashed black lines highlight the temperature evolution of the three most pronounced resistance spikes. Triangles and black dots mark the corresponding peak positions in Fig. 3a. Red dotted lines indicate multiple superconducting transitions.

ZrNCl[26] and $d_{SC} = 2.65$ nm for the monolayer WTe$_2$[34]. Superconducting fluctuations are expected to be substantially weakened in our system, because the superconducting puddles in a single layer can be bridged by the puddles in the neighboring layer, as illustrated in the inset of Fig. 4b.

## Discussion

To understand the emergence of AM in a relatively thick sample, we put together the data obtained with the two methods—(1) sweeping $B$ at fixed $T$ and (2) varying $T$ at fixed $B$—in Fig. 4c and Supplementary Fig. 17. Here we subtract the magnetoresistance by a smoothened background (details are given in Supplementary Fig. 16) and plot the difference as a function of temperature and magnetic field. In comparison, we mark by empty circles in Fig. 4c the temperature points where the resistance with decreasing temperature deviates from the thermally activated behavior (defined in Fig. 4a). They are conventionally defined as the onset temperature of AM[26,35–37]. We further employ two more criteria to define the AM phase. First, we apply the method used before in liquid-gated ultrathin TiSe$_2$ by defining $T_{AM}$ as the crossing point between linear extrapolations of: (1) the thermally activated behavior and (2) the resistance plateau at low

temperatures, as exemplified in the left panel of Fig. 4a[27]. Second, the onset temperature for the resistance plateau is marked as $T_{AM0}$. As shown in Fig. 4c, the resistance oscillations are most pronounced in the region below $T_{AM}$ and the red and blue stripes extend up to the onset of the AM.

The coexistence of the resistance oscillations and the AM gives us some clue on the mechanism for the AM in our multi-layer system. The spontaneously formed periodic structure, manifested by the resistance oscillations, may align the superconducting puddles of the individual layers in the vertical direction, as schematically illustrated in the inset of Fig. 4b. The superconducting fluctuations are therefore enhanced, in a similar manner to patterning a relatively thick superconductor into periodically aligned arrays[30–32], allowing the emergence of AM. Our work therefore sheds a distinct perspective on the link between the AM state and superconducting fluctuations.

To summarize, we realize superconductivity with a maximal $T_{c0}$ of 1.5 K in lithium-intercalated TiSe$_2$ flakes and observe two intriguing phenomena that were previously exclusive to the ultrathin case: (1) magnetoresistance oscillations around the superconducting transition regime and (2) AM phase. Their coexistence in the same temperature, magnetic field, and doping ranges suggests an intimate link between them. Notably, the

periodic structure, interpreted from the resistance oscillations, hosts unique doping and temperature dependences. These behaviors expose details in the evolution of the electronic ordering in $TiSe_2$ and pose further constraints on a comprehensive theoretical model. In comparison to the ionic liquid-gated ultrathin $TiSe_2$, which was the only system hosting (1) and (2) before, our $Li_xTiSe_2$ is compatible with surface-sensitive techniques, as demonstrated already by our in situ AFM study. This advantage opens up further opportunities in addressing the exotic physics in $TiSe_2$. It allows for the real-space investigation of the electronic ordering by using techniques such as scanning tunneling microscopy. Furthermore, we demonstrate that the periodic domain structure can emerge in a rather thick sample, which welcomes future experiments with X-ray diffractions.

## Methods

The $TiSe_2$ crystals are mechanically exfoliated by scotch tape in a glove box with low concentration of $H_2O$ and $O_2$ (<0.1 p.p.m.) and dry transferred onto the solid ion conductor substrates with pre-patterned electrodes ($AlO_x$/Ti/Au: 10/5/30 nm). The $AlO_x$ layer prevents lithium ions from intercalating into the metal electrodes. The devices are then wire-bonded and loaded into a closed-cycle system (Oxford Instruments TeslatronPT) with a $^3$He insert. We install room temperature π filters (two 35 nF capacitors and one 220 μH inductor) to reduce the high-frequency noises. The resistance and Hall measurements are performed with the standard lock-in technique. The excitation current is chosen to be 1 μA (13 Hz), to avoid current-induced heating (Supplementary Fig. 9). The back-gate voltage is applied with a DC source-meter (Keithley 2400).

## Data availability

The data represented in Figs. 1–4 in this study have been deposit in: D. Zhang, Data for "Superconductivity and related phenomena in lithium intercalated $TiSe_2$," https://doi.org/10.7910/DVN/KRNGH3, Harvard Dataverse (2020). All other data that support the plots within this paper are available from the corresponding author upon reasonable request.

## Code availability

The computer code used for data analysis is available upon request from the corresponding author.

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

## Acknowledgements

We thank Haiwen Liu, Yijin Zhang, Joseph Falson, and Benedikt Friess for fruitful discussions and proofreading the manuscript. This work is financially supported by the National Natural Science Foundation of China (grant numbers 51788104, 11790311 and 11922409), the Ministry of Science and Technology of China (2017YFA0302902 and 2017YFA0304600), and the Beijing Advanced Innovation Center for Future Chip (ICFC).

## Author contributions

D.Z. and L.Y. initiated the project. M.L. fabricated the samples and did transport measurements. H.W., Y.Z., R.S., and H.Z. assisted in various stages of the experiment. M.L. and D.Z. analyzed the data with M.R.'s assistance. M.L. and D.Z. wrote the paper with the input from L.Y. and Q.-K.X. All authors discussed the results and commented on the manuscript.

## Competing interests

The authors declare no competing interests.
