## [Peer Review File · Nature Communications]

Editorial Note: Parts of this peer review file have been redacted as indicated to remove third-party material where no permission to publish could be obtained.

REVIEWER COMMENTS

Reviewer #1 (Remarks to the Author):

In the present manuscript entitled "Coexistence of resistance oscillations and the anomalous 2 metal phase in a lithium intercalated TiSe₂ superconductor", Liao and their colleagues report on transport properties in Li-intercalated TiSe₂. The main finding is that the periodic oscillation appears in magnetoresistance and the periodicity is carrier density-dependent in a "bulk" crystal (50 nm thick). In addition, they found that the mean free path increases with doping, which is in contrast to the previous similar work on ionic-gated ultra-thin TiSe₂. Such a forming periodic domain could be an alternative explanation for the quantum metallic state, which is frequently observed in 2D superconductors and has been usually ascribed to the quantum fluctuation.

The manuscript is well written, and the experimental results are convincing and of interest in the field of 2D superconductors. Therefore, I recommend publication in Nature Communications after the authors address the following comment.

1) It would be interesting to explore the wide range of thickness dependence, say 5 nm – 20 nm, of this periodic oscillation, is magnetoresistance. Also, would this phenomenon be different if you use other ionic liquids (e.g. DEMA-TFSI)?

Reviewer #2 (Remarks to the Author):

The study by Liao et al. investigates the so-called anomalous metal (AM) "phase" of Li_xTiSe₂, specifically looking at how it connects to "resistance oscillations." The basic finding, as far as I can discern, is that the appearance of the AM correlates with resistance oscillations. The latter are connected to superconducting (SC) "domains" separated by regions of CDW.

I do not have too many technical complaints about the study, but the conclusions seem a little weak. 1. The proposal that there are SC domains is not well established. First of all, have the authors performed an angle dependence of the resistance oscillations? This will presumably convolve H_{c2} anisotropy if they are truly SC domains. 2. The AM does not seem well defined - what are the key characteristics of it other than showing up as a plateau? 3. The connection between SC fluctuations and AM that the authors claim to have found is really very ambiguous. What connection exactly have they found? Do they mean spatial fluctuations? 4. Please forgive my utter ignorance, but why is any of this surprising? Inhomogeneous SC will inevitably lead to some lower resistance state whose percolation path is incomplete, leading to something that looks like an AM. 5. Do the authors have a microscopic picture for the h/2e oscillations? Are there SC vortices the size of the domains with edge states that skip around the surrounding normal or CDW phase?

While this paper certainly deserves to be published somewhere, it isn't at all clear to me that it meets the high standards of Nature Communications.

Reviewer #3 (Remarks to the Author):

The paper by M. Liao et al. reports on superconducting transitions and magnetic field-induced

resistance oscillations observed for 50 nm-thick lithium intercalated TiSe₂. For samples with carrier density $\sim 10^{16} \text{ cm}^{-2}$, the CDW phase in the non-doped TiSe₂ is partially suppressed and a superconducting transition appears with a maximum T_c of 1.5 K. The magnetoresistance oscillations appear in the optimum doping regime, which is attributed to the Little-Park effect caused by periodic formation of superconducting domains. The temperature and doping dependences of the oscillation are different from those reported for electric-field doped ultrathin TiSe₂ samples, where the formation of CDW domains was indicated. The so-called anomalous metal phase is identified from the saturation of resistance decrease at the lowest temperatures, which suggests that strong fluctuations caused by the periodic superconducting domains play an important role. The paper is timely and includes interesting physics. Nevertheless, I have several concerns in terms of interpretation of the results, which is central to the novelty of the paper. The authors are requested to answer the following questions and criticisms.

1) p.5, ll.150-158 "Here, instead, we speculate that the superconducting regions, instead of CDW, form domains..... because the superconducting domains expand at an increased doping level (inset to Fig. 3h).":

Here the authors claim that the periodic pattern consists of superconducting domains rather than of CDW. However, I suspect that it should be attributed to a CDW formation as was reported in Ref. [6] based on the following reasons.

A) The Little-Parks effect can naturally explain the result of Ref. [6], since the superconducting region consists of a loop-and-hole structure (i.e. multiply connected); the area of the hole (A) determines the period of the Little-Parks oscillation (Δ_B) through a relation $\Delta_B \cdot A = h/2e$. By contrast, in the model proposed here, each superconducting region has no such structure (i.e. simply connected). This means that there should be no well-defined magnetoresistance oscillation. The clear oscillation of R as a function of B (Figs. 3d and 3e) strongly suggests the presence of a loop-and-hole structure due to the CDW formation.

B) The periodic domain formation of CDW and the emergence of superconductivity in the domain walls have been reasonably explained in Ref. [6]. By contrast, spontaneous formation of periodic superconducting regions (without the help of CDW) is hard to imagine. Note that strong spatial modulation of superconducting order parameter (to the extent that superconductivity is locally destroyed) costs a lot of energy and should be unfavorable.

C) In the course of field-induced transition, increase in magnetic field should weaken superconductivity and leads to shrinkage of the superconducting domain size, according to the authors' argument. However, well-defined domain size L was identified from the periodic oscillation, indicating that the relevant phenomenon is not superconductivity but rather CDW.

2) Figs. 4a,4b:

How did the authors determine the points where the temperature dependence of resistance deviates from the thermal activation behavior? Since this transition is a continuous crossover, it is not straightforward to separate the two temperature regions. To me, some of the points seems to be intentionally adjusted so that it fits the "stripe features" in the phase diagram of Fig.4d (for example, the data for 0.05T of #4 and #5). The authors should show a systematic and objective criterion for the judgement.

3) p. 6, "However, interlayer coupling in our system—a thick stack of TiSe₂ layers (Fig. 4e)—is expected to weaken such fluctuations ...As a result, the AM only shows up when the periodic domains—manifested by the resistance oscillations—emerge"

The authors claim that the interlayer coupling is expected to weaken the superconducting fluctuations. But this is contradictory to their finding that the angular dependence of B_{c2} follows the 2D Tinkham formula and rather than an anisotropic Ginzburg-Landau model and their conclusion that their samples consists of a stack of (isolated) 2D superconductors. Indeed, even for a doping regime where the magnetoresistance oscillation is not observed (Figs. S10 and S11), deviations from the thermal activation behavior are clearly observed. Although the authors' claim is "the saturation behavior is absent" for these samples, I do not agree with this statement; the temperature dependences are qualitatively the same for under-/optimum-/over- doped samples. This strongly indicates that the anomalous metal phase is not directly related to the periodic domain formation in the samples.

Reviewer #1 (Remarks to the Author):

In the present manuscript entitled “Coexistence of resistance oscillations and the anomalous 2 metal phase in a lithium intercalated TiSe₂ superconductor”, Liao and their colleagues report on transport properties in Li-intercalated TiSe₂. The main finding is that the periodic oscillation appears in magnetoresistance and the periodicity is carrier density-dependent in a “bulk” crystal (50 nm thick). In addition, they found that the mean free path increases with doping, which is in contrast to the previous similar work on ionic-gated ultra-thin TiSe₂. Such a forming periodic domain could be an alternative explanation for the quantum metallic state, which is frequently observed in 2D superconductors and has been usually ascribed to the quantum fluctuation.

The manuscript is well written, and the experimental results are convincing and of interest in the field of 2D superconductors. Therefore, I recommend publication in Nature Communications after the authors address the following comment.

[REPLY] We would like to thank the reviewer for the high recognition of our work and for recommending our paper for publication.

1) It would be interesting to explore the wide range of thickness dependence, say 5 nm – 20 nm, of this periodic oscillation, is magnetoresistance. Also, would this phenomenon be different if you use other ionic liquids (e.g. DEME-TFSI)?

[REPLY] Following the nice suggestion of the reviewer, we have carried out further investigation on thinner samples. The superconductivity becomes less pronounced as the sample thickness decreases. Typical results are shown in Fig. R1 below. The two pristine TiSe₂ flakes have thicknesses of 10 nm and 15 nm, respectively. Both samples show a dome-like superconducting phase diagram as a function of lithium intercalation. However, the optimal superconducting transition temperature here is about 0.8 K, which is about half of that in the 50 nm thick samples. The weakened superconductivity may be related to the reduced total density of states of the material and the stronger disorder effect. In Fig. R2, we show that the carrier mobility of the thinner samples is much lower than that of the thick ones. Figure R3 shows the magneto-resistance at the base temperature (0.25 K) for the 15 nm thick sample. There exist no clear oscillatory features. It could be due to the weakened superconductivity. Furthermore, the stronger disorder in thinner samples can be detrimental to the spontaneous formation of a periodic structure.

As for the dependence of the resistance oscillations on the gating material, we note that the previous report [Nature 529, 185 (2016), ref. (6)] employed the ionic liquid, rather than the ionic solid by us. The resistance oscillations there indeed behaved differently. The oscillation period was temperature independent and became larger as the carrier density increased. These phenomena are sharply different from what we observed in lithium intercalated TiSe₂ films. In the revised manuscript, we provide more discussions to draw the readers' attention to these differences.

Fig. R1 Temperature dependent resistances of two TiSe_2 flakes under lithium intercalation. The carrier density at each intercalated state is calculated from the Hall measurement at 1.6 K. Panels **a** and **b** show the resistance curves in a large temperature range while panels **c** and **d** show the same data set but in the low temperature regime only. The black arrows mark the superconducting onset temperature ($90\%R_n$). The arrow marks T_{c0} for the trace at optimal doping.

Fig. R2 Low temperature carrier mobility values of Li_xTiSe_2 samples as a function of single-layer carrier density. The two relatively thin samples (R1 and R2) have lower mobility than those of thick samples (S1 and S2).

Fig. R3 Magneto-resistance of sample R1 (15 nm) at different doping levels. All curves are obtained at 0.25 K. No prominent resistance oscillations can be identified.

Reviewer #2 (Remarks to the Author):

The study by Liao et al. investigates the so-called anomalous metal (AM) "phase" of Li_xTiSe_2 , specifically looking at how it connects to "resistance oscillations." The basic finding, as far as I can discern, is that the appearance of the AM correlates with resistance oscillations. The latter are connected to superconducting (SC) "domains" separated by regions of CDW.

I do not have too many technical complaints about the study, but the conclusions seem a little weak.

[REPLY] We would like to thank the reviewer for the positive remark on our experiments. Following the important suggestion of the reviewer, we have carried out further experiments and more data analysis. We have also significantly revised our manuscript to improve its structuring and highlight the important findings.

1. The proposal that there are SC domains is not well established. First of all, have the authors performed an angle dependence of the resistance oscillations? This will presumably convolve H_{c2} anisotropy if they are truly SC domains.

[REPLY] Following the nice suggestion of the reviewer, we have carried out further experiments on the angular dependence. We observe that the resistance oscillations only depend on the magnetic field that is perpendicular to the sample plane. It indicates that the periodic structure is two-dimensional. We believe this finding substantially strengthens our work since no prior work has revealed this 2D property. This important piece of information has been added to the supplementary information in section 6 as Fig. S14. We have also described these results in the main text (first paragraph of the section titled "Magneto-resistance oscillations").

In the revised manuscript, we have substantially revised the texts and attributed the resistance oscillations to the formation of CDW domains. More explanations are provided in the answer to point 5.

2. The AM does not seem well defined - what are the key characteristics of it other than showing up as a plateau?

[REPLY] The resistance plateau at temperature below the superconducting transition, with a resistance value that is orders of magnitude lower than the normal state resistance, is the defining characteristic of the anomalous metal phase. We provide further explanation on why this behavior is exotic in the answer to point 4.

Apart from the resistance plateau, the large magneto-resistance serves as a characteristic feature for the anomalous metal phase, as pointed out in [Science **350**, 409 (2015), ref. (26)].

In the revised manuscript, we have added the corresponding analysis on the magneto-resistance of the anomalous metal phase. This is mentioned in the main text and details are given in the supplementary information as section 7. We show that the magneto-resistance can be well fit by the quantum creeping model of the anomalous metal.

3. The connection between SC fluctuations and AM that the authors claim to have found is really very ambiguous. What connection exactly have they found? Do they mean spatial fluctuations?

[REPLY] We have significantly revised our discussion to clarify this point. In general, we wish to provide an explanation to the seeming contradiction that the anomalous metal phase occurs in a relatively thick superconductor. This is counter-intuitive because the anomalous metal phase is widely believed to be related to the strong superconducting phase fluctuations in two-dimensional space [see ref. (33), the fluctuations occur spatially]. The superconducting thickness of our sample ($d_{SC}=20$ nm), however, is orders of magnitude larger than those of the ultrathin superconductors ($d_{SC}=1.5$ nm) such that the corresponding superconducting fluctuations in our samples should be substantially weakened. We believe the formation of a periodic structure in our sample strongly enhances the superconducting fluctuations, which is similar to that achieved by patterning the a relatively thick superconductor into periodic arrays, such that the anomalous metal phase emerges. A schematic drawing of this scenario has been added in the inset of Fig. 4b.

4. Please forgive my utter ignorance, but why is any of this surprising? Inhomogeneous SC will inevitably lead to some lower resistance state whose percolation path is incomplete, leading to something that looks like an AM.

[REPLY] The important findings of our work include: (1) the resistance oscillations; (2) the anomalous metal phase; (3) the overlap of the phase diagrams of (1) and (2). We believe the resistance oscillations are surprising because it indicates that there exists a spontaneously emerged periodic structure. As for points (2) and (3), we have provided some explanations on their uniqueness in the previous point. Here we wish to discuss the role of percolation, which the reviewer pointed out.

The reviewer is correct that percolation plays an important role in giving rise to the anomalous metal. However, we emphasize that a classical percolation model cannot explain the result. The following argument has been added to the main text:

“We further note that the resistance plateau cannot be explained by a classical percolation model, in which the superconducting regions are separated by the normal state regions. To account for the remnant resistance of $4\%R_n$ for the trace at 0.075 T in the left panel of Fig. 4a, the fraction of normal state regions has a maximal length of 200 nm, because the distance between the two voltage probes for measuring the resistance is 5 μm . On the other

hand, the normal state coherence length $\xi_n = \sqrt{\hbar D/k_B T}$ exceeds 200 nm at $T < 0.3$ K (here $D = v_F l/2$ is the diffusion constant and v_F is the Fermi velocity. We estimate v_F to

be 3×10^5 m/s based on the band mass reported in [8]). In other words, the normal state region can be fully proximitized by the superconducting region such that the resistance should drop down to zero at $T < 0.3$ K.”

Instead, a quantum percolation model is required to explain the anomalous metal phase. For example, the quantum creeping of vortices, which gives rise to dissipation below the superconducting transition, is considered in one theoretical model for the anomalous metal [Science **350**, 409 (2015), ref. (26), Phys. Rev. Lett. **80**, 3352 (1998), ref. (7) in the supplementary information]. It involves the tunneling of vortices across the percolated superconducting puddles. However, the quantum model seems to be under-developed, as we quote the following statement in a recent review paper [Rev. Mod. Phys. **91**, 011002 (2019), ref. (33)]: “We argue that there is currently no satisfactory theory of anomalous metals that accounts for the full set of key experimental facts, in particular, the robustness of the anomalous metallic state”. In this regard, we believe that realizing the anomalous metal in a relatively thick sample of TiSe_2 provides a unique platform for further investigating the theoretical models.

5. Do the authors have a microscopic picture for the $h/2e$ oscillations? Are there SC vortices the size of the domains with edge states that skip around the surrounding normal or CDW phase?

[REPLY] In the revised manuscript, we provide more discussions on the microscopic model, which is based on the scenario proposed in [Nature **529**, 185-189 (2016), ref. (6)]. The detailed discussions constitute the second paragraph of the section titled “Magneto-resistance oscillations”. For convenience, we attach a clip of the figure from the reference below (Fig. R4). There, the periodic structure consists of domains of commensurate CDW (CCDW) and domain walls of incommensurate CDW (ICDW). At low temperature, superconductivity emerges in the ICDW region whereas the CCDW region remains non-superconducting. The system therefore forms loop-and-hole structures such that the Little-Parks effect appears.

[Redacted]

Fig. R4 Phase diagram of ionic liquid gated ultrathin TiSe_2 flake, as taken from Nature **529**, 185-189 (2016). The inset illustrates the periodic structure, which consists of domains of commensurate CDW

and domain walls of incommensurate CDW.

We emphasize that although the theoretical model is not new, our experimental results are valuable because: (1) we observe distinctly different doping and temperature dependences of the oscillations; (2) technically, we realize the resistance oscillations in a thick sample with free surface, allowing future experiments such as scanning probe microscopy.

While this paper certainly deserves to be published somewhere, it isn't at all clear to me that it meets the high standards of Nature Communicaitons.

[REPLY] We would like to take this chance to emphasize the important findings of our work.

Apart from the dome-like phase diagram of superconductivity we realized by a solid-state gating technique in TiSe_2 , we would like to emphasize the three major findings in this superconductor:

(1) We observe resistance oscillations in Li_xTiSe_2 , indicating the spontaneous formation of a periodic structure that modulates superconductivity. Although a study on ionic liquid gated TiSe_2 has revealed a similar behavior [*Nature* **529**, 185-189 (2016), ref. (6)], we note that that report has so far remained to be the only case. More importantly, we observe completely different doping and temperature dependences of the oscillations than those reported before. These results substantially broaden our understanding of the spontaneously formed periodic structure.

(2) We observe resistance plateaus at low temperatures and small magnetic fields—a key characteristic of the anomalous metal phase. In contrast to all the previous reports, the anomalous metal phase here manifests itself in a rather thick sample, as we show in Fig. 4b.

(3) We observe a close correlation between the anomalous metal phase and the resistance oscillations, as the reviewer kindly pointed out. Although this third point is important, we believe the other two points also represent novel physics and distinguish our work from all previous works.

Reviewer #3 (Remarks to the Author):

The paper by M. Liao et al. reports on superconducting transitions and magnetic field-induced resistance oscillations observed for 50 nm-thick lithium intercalated TiSe₂. For samples with carrier density $\sim 10^{16} \text{ cm}^{-2}$, the CDW phase in the non-doped TiSe₂ is partially suppressed and a superconducting transition appears with a maximum T_c of 1.5 K. The magnetoresistance oscillations appear in the optimum doping regime, which is attributed to the Little-Park effect caused by periodic formation of superconducting domains. The temperature and doping dependences of the oscillation are different from those reported for electric-field doped ultrathin TiSe₂ samples, where the formation of CDW domains was indicated. The so-called anomalous metal phase is identified from the saturation of resistance decrease at the lowest temperatures, which suggests that strong fluctuations caused by the periodic superconducting domains play an important role. The paper is timely and includes interesting physics. Nevertheless, I have several concerns in terms of interpretation of the results, which is central to the novelty of the paper. The authors are requested to answer the following questions and criticisms.

[REPLY] We would like to thank the reviewer for the nice summary of our work and for the positive remarks. We provide a point-by-point answer below.

1) p.5, ll.150-158 “Here, instead, we speculate that the superconducting regions, instead of CDW, form domains..... because the superconducting domains expand at an increased doping level (inset to Fig. 3h).”:

Here the authors claim that the periodic pattern consists of superconducting domains rather than of CDW. However, I suspect that it should be attributed to a CDW formation as was reported in Ref. [6] based on the following reasons.

A) The Little-Parks effect can naturally explain the result of Ref. [6], since the superconducting region consists of a loop-and-hole structure (i.e. multiply connected); the area of the hole (A) determines the period of the Little-Parks oscillation (Δ_B) through a relation $\Delta_B \cdot A = h/2e$. By contrast, in the model proposed here, each superconducting region has no such structure (i.e. simply connected). This means that there should be no well-defined magnetoresistance oscillation. The clear oscillation of R as a function of B (Figs. 3d and 3e) strongly suggests the presence of a loop-and-hole structure due to the CDW formation.

B) The periodic domain formation of CDW and the emergence of superconductivity in the domain walls have been reasonably explained in Ref. [6]. By contrast, spontaneous formation of periodic superconducting regions (without the help of CDW) is hard to imagine. Note that strong spatial modulation of superconducting order parameter (to the extent that superconductivity is locally destroyed) costs a lot of energy and should be unfavorable.

C) In the course of field-induced transition, increase in magnetic field should weaken superconductivity and leads to shrinkage of the superconducting domain size, according to the authors' argument. However, well-defined domain size L was identified from the periodic oscillation, indicating that the relevant phenomenon is not superconductivity but rather CDW.

[REPLY] We would like to thank the reviewer for the extensive explanation. The reviewer is correct that the periodic structure in our case should be caused by CDW. More specifically, the periodic domain is occupied by the commensurate CDW (CCDW) while the domain boundary possesses the incommensurate CDW (ICDW). At low temperatures, the ICDW region becomes superconducting. It is this superconducting region that we described as the periodic superconducting regions.

In the previous version, however, we assigned the domains to be ICDW while the domain boundary to be CCDW, which was opposite to the situation in ref. [6]. We speculated that this scenario was favored in our case because of the dominant ICDW over CCDW in the optimal to nearly over-doped regime. However, we have realized that the experimental data so far cannot distinguish between the two scenarios. The description in the main text has been therefore significantly revised and we essentially follow the model in ref. [6]. Still, we emphasize that there exist clear distinctions between our results and those of ref. [6]: different doping and temperature dependences. We believe these differences greatly broaden the scope of spontaneously formed periodic structure and pave an important step toward a quantitative understanding of this bizarre phenomenon.

Furthermore, in the revised manuscript, we have managed, for the first time, to demonstrate the two-dimensional nature of the periodic structure by carrying out angular dependent study of the oscillations. This important piece of information has been added to the supplementary information in section 6 as Fig. S14. We have also described these results in the main text (first paragraph of the section titled "Magneto-resistance oscillations").

2) Figs. 4a,4b:

How did the authors determine the points where the temperature dependence of resistance deviates from the thermal activation behavior? Since this transition is a continuous crossover, it is not straightforward to separate the two temperature regions. To me, some of the points seems to be intentionally adjusted so that it fits the "stripe features" in the phase diagram of Fig.4d (for example, the data for 0.05T of #4 and #5). The authors should show a systematic and objective criterion for the judgement.

[REPLY] In the revised manuscript, we have followed the suggestion of the reviewer and applied a rigorous criterion for defining the deviation point. It is defined as the point where the resistance deviates from the thermally activated linear behavior by 5% with decreasing temperature. We have specified this definition in the figure caption of Fig. 4. Furthermore, we have included two other criteria to define the anomalous metal phase. The following texts have been added to the main text:

"First, we apply the method used before in liquid gated ultrathin TiSe₂ by defining T_{AM} as

the crossing point between linear extrapolations of: (1) the thermally activated behavior and (2) the resistance plateau at low temperatures, as exemplified in the left panel of Fig. 4a.”

We provide more details about the procedure for defining the temperature points in the figure caption of Fig. 4:

“The trace at 0.075 T in the left panel helps illustrate the method for further defining the anomalous metal. The dotted horizontal line represents the resistance plateau, which is obtained by averaging R_s from 0.25 to 0.35 K. The standard deviation of R_s in this temperature window is δR_s . T_{AM} marks the crossing point between the dashed and horizontal lines. T_{AM0} marks the point at which R_s first exceeds the horizontal line by $10\delta R_s$ when T increases from 0.25 K.”

3) p. 6, “However, interlayer coupling in our system—a thick stack of TiSe₂ layers (Fig. 4e)—is expected to weaken such fluctuations ...As a result, the AM only shows up when the periodic domains—manifested by the resistance oscillations—emerge”

The authors claim that the interlayer coupling is expected to weaken the superconducting fluctuations. But this is contradictory to their finding that the angular dependence of Bc₂ follows the 2D Tinkham formula and rather than an anisotropic Ginzburg-Landau model and their conclusion that their samples consists of a stack of (isolated) 2D superconductors.

[REPLY] We have revised the discussion part to clarify this point. Although the angular dependence of Li_xTiSe₂ follows the 2D Tinkham formula, we note that the extracted superconducting thickness d_{SC} is around 20 nm, which is thinner than the sample thickness but much thicker than one atomic layer of TiSe₂. We therefore conclude that our system consists of a stack of 2D superconductors. Despite the fact that superconducting fluctuations can still separate each 2D plane into superconducting puddles, these puddles may couple vertically via the Josephson coupling, as we schematically illustrate in the inset of Fig. 4b of the revised manuscript. We believe this vertical coupling could suppress the fluctuation effect and the anomalous metal is not expected to occur.

Furthermore, we note that most of the previous cases that reported the anomalous metal phase occurred in ultrathin 2D superconductors with d_{SC} that was one order of magnitude smaller than that in our case. For example, ionic liquid gated ZrNCl possessed d_{SC} of 1.8 nm [*Science* **350**, 409-413 (2015), ref. (26)] and monolayer WTe₂ had $d_{SC} = 2.65$ nm [*Science* **362**, 922-925 (2018), ref. (34)]. In NbSe₂, although the origin of the apparent anomalous metal behavior is under debate [*Nat. Phys.* **12**, 208-212 (2016), ref. (28). *Sci. Adv.* **5**, eaau3826 (2019), ref. (29)], the extracted d_{SC} is 3.4 nm. These ultrathin superconducting systems are close to the ideal 2D situation such that they are prone to superconducting fluctuations.

We have reorganized the main text and discuss the possible reason for the AM in our thick sample in the last section. We propose the following mechanism:

“The spontaneously formed periodic structure, manifested by the resistance oscillations,

aligns the superconducting puddles of the individual layers in the vertical direction, as schematically illustrated in the inset of Fig. 4b. The superconducting fluctuations are therefore enhanced, in a similar fashion to patterning a relatively thick superconductor into periodically aligned arrays [30-32], allowing the emergence of AM.”

Indeed, even for a doping regime where the magnetoresistance oscillation is not observed (Figs. S10 and S11), deviations from the thermal activation behavior are clearly observed. Although the authors’ claim is “the saturation behavior is absent” for these samples, I do not agree with this statement; the temperature dependences are qualitatively the same for under-/optimum-/over- doped samples. This strongly indicates that the anomalous metal phase is not directly related to the periodic domain formation in the samples.

[REPLY] We have edited Fig. 4a to highlight the qualitative difference between the curves for the optimal doped and those for the over-doped. In the nearly optimal doped to slightly over-doped cases (left and middle panels), the resistance shows a plateau that can span from around $1/T = 2.5 \text{ K}^{-1}$ to 4 K^{-1} . In contrast, the resistance keeps decreasing with increasing $1/T$ in the over-doped case (right panel), showing no indication of a resistance plateau. To be more precise about the feature that is different, we have changed the wording from “the saturation behavior” to “the resistance plateau”.

We mainly use the doping dependent study to rule out insufficient cooling as the origin for the anomalous metal phase. This has been stated in the revised manuscript as: “Insufficient cooling can be immediately ruled out because otherwise the same technical issue should be present at different doping levels. In the over-doped regime (right panel of Fig. 4a), however, the resistance plateau is absent. Instead, the resistance shows a continuously decreasing trend down to the base temperature.”

We remark that the bending points at around $1/T = 1$ and 2 K^{-1} for the curves in the over-doped situation (right panel of Fig. 4a) stem from multiple superconducting transitions, as can be better appreciated by the same data set shown in the revised Fig. 2b (#6). We therefore refrain from using them as indicators for the onset of the anomalous metal. We also note that the data in Fig. S11 were obtained without the installation of filters. The continuous bending of the curves in Fig. S11 may therefore be caused by the radio frequency noises. Nevertheless, the resistance keeps decreasing with decreasing temperature (increasing $1/T$), which again rules out insufficient cooling as the reason for the resistance plateau at optimal doping.

REVIEWERS' COMMENTS

Reviewer #1 (Remarks to the Author):

The authors correctly addressed my concern, and therefore I would fully recommend publication in Nature Communications.

Reviewer #3 (Remarks to the Author):

All the necessary revisions have been made regarding my comments. The responses to the other reviewers seem satisfactory to me. Now I recommend its publication in Nature Communications.

Reviewer #1 (Remarks to the Author):

The authors correctly addressed my concern, and therefore I would fully recommend publication in Nature Communications.

[REPLY] We thank the referee for recommending our paper for publication.

Reviewer #3 (Remarks to the Author):

All the necessary revisions have been made regarding my comments. The responses to the other reviewers seem satisfactory to me. Now I recommend its publication in Nature Communications.

[REPLY] We thank the referee for recommending our paper for publication.